# Expanded Archaeal Genomes Shed New Light on the Evolution of Isoprenoid Biosynthesis

**DOI:** 10.3390/microorganisms12040707

**Published:** 2024-03-30

**Authors:** Pengfei Zhu, Jialin Hou, Yixuan Xiong, Ruize Xie, Yinzhao Wang, Fengping Wang

**Affiliations:** 1Key Laboratory of Polar Ecosystem and Climate Change, Ministry of Education, School of Oceanography, Shanghai Jiao Tong University, Shanghai 200240, China; zhupengfei@sjtu.edu.cn (P.Z.); houjialin6@sjtu.edu.cn (J.H.); xiongyx83@sjtu.edu.cn (Y.X.); xieruize@sjtu.edu.cn (R.X.); 2State Key Laboratory of Microbial Metabolism, School of Life Sciences and Biotechnology, Shanghai Jiao Tong University, Shanghai 200240, China; wyz@sjtu.edu.cn; 3Southern Marine Science and Engineering, Guangdong Laboratory (Zhuhai), Zhuhai 519080, China

**Keywords:** isoprenoids, mevalonate pathway, LUCA, Asgard archaea, eukaryogenesis

## Abstract

Isoprenoids and their derivatives, essential for all cellular life on Earth, are particularly crucial in archaeal membrane lipids, suggesting that their biosynthesis pathways have ancient origins and play pivotal roles in the evolution of early life. Despite all eukaryotes, archaea, and a few bacterial lineages being known to exclusively use the mevalonate (MVA) pathway to synthesize isoprenoids, the origin and evolutionary trajectory of the MVA pathway remain controversial. Here, we conducted a thorough comparison and phylogenetic analysis of key enzymes across the four types of MVA pathway, with the particular inclusion of metagenome assembled genomes (MAGs) from uncultivated archaea. Our findings support an archaeal origin of the MVA pathway, likely postdating the divergence of Bacteria and Archaea from the Last Universal Common Ancestor (LUCA), thus implying the LUCA’s enzymatic inability for isoprenoid biosynthesis. Notably, the Asgard archaea are implicated in playing central roles in the evolution of the MVA pathway, serving not only as putative ancestors of the eukaryote- and *Thermoplasma*-type routes, but also as crucial mediators in the gene transfer to eukaryotes, possibly during eukaryogenesis. Overall, this study advances our understanding of the origin and evolutionary history of the MVA pathway, providing unique insights into the lipid divide and the evolution of early life.

## 1. Introduction

As the largest family of organic compounds in nature, isoprenoids and their derivatives contain over 50,000 molecules and play essential physiological roles in bacterial, archaeal, and eukaryotic cells (including human), such as the biosynthesis of archaeal membrane lipids, the electron transport chain, and the photosynthesis system [1,2]. The carbon skeleton of all isoprenoids is derived from the successive condensations of two C_5_ precursors: isopentenyl diphosphate (IPP) and its isomer dimethylallyl diphosphate (DMAPP), both of which are biosynthesized through either the methylerythritol 4-phosphate (MEP) pathway or the mevalonate (MVA) pathway [1,3]. Among them, most bacteria and plastid-bearing eukaryotes (e.g., plants, green algae, and apicomplexa) possess the MEP pathway [4,5,6], whereas the MVA pathway is universally employed by all eukaryotes and archaea, as well as a few bacterial species [3,7]. Therefore, the physiological essentiality and ubiquitous distribution suggest the crucial roles and an ancient origin of the MVA pathway in the evolution of early life on Earth.

The classic MVA pathway firstly discovered in yeasts is possessed by all eukaryotic life and is thus referred to as the eukaryote-type MVA pathway. However, the complete gene repertoire of the classic pathway has only been identified in the thermoacidophilic archaeon *Sulfolobales* [7,8] and some DPANN archaea [9,10] as the sole isoprenoid biosynthesis pathway. In addition, three alternative variants of the MVA pathway have been discovered in the domain Archaea so far: (1) the haloarchaea-type MVA pathway that was discovered in the haloarchaeal archaea and green non-sulfur bacteria (phylum *Chloroflexota*) [11,12,13], which are not only commonly found in haloarchaea for carotenogenesis [14] but also detected in some thermophilic archaea from *Thermoplasmata* [10]; (2) the *Thermoplasma*-type pathway that is specific for *Thermoplasmatales* [15,16,17] and some *Micrarchaeota* [18]; (3) and the most recently discovered archaea-type MVA pathway, which is supposed to be conserved in the majority of archaea except those lineages mentioned above [19]. Biochemically, all the four types share the initial and the last enzymatic steps, including the conversions from acetoacetyl-CoA to MVA via 3-hydroxy-3-methylglutaryl-CoA synthase (HMGS) and 3-hydroxy-3-methylglutaryl-CoA reductases (HMGR), and the isomerization from IPP to DMAPP via the IPP/DMAPP isomerase (IDI) (Figure 1a). Except the *Thermoplasma*-type pathway found in a handful of thermophilic archaea, all the other three types of MVA pathway encode the mevalonate kinase (MVK) that catalyzes phosphorylation of MVA. Moreover, the key distinction among the four MVA pathways is the synthetic route from MVA to IPP, which is catalyzed by specific enzymes through distinct substrates (Figure 1a), reflecting the metabolic diversity of isoprenoid biosynthesis in the domain Archaea.

Given the indispensable roles of isoprenoids in all domains of life, especially archaeal membrane lipids, the isoprenoid synthesis pathway thus is inferred to have an ancient origin [3,20]. The MVA pathways were previously thought to have independent origins in the domains Eukarya and Archaea, respectively, and were subsequently acquired by a few bacterial species via multiple independent horizontal gene transfer (HGT) events [3,21]. Later, the finding of the eukaryote-type pathway in more phyla of bacteria and the superphylum Candidate Phyla Radiation (CPR) further raised the possibility that the eukaryote-type MVA pathway as an ancient type may have emerged instead in the bacterial ancestor HGT, suggesting the origin of the MVA pathway in the LUCA [10,22]. Nonetheless, some researchers suggest that the phylogenetic position of CPR, upon which the inference relies, may not be situated at the root of the bacterial tree of life [23,24]. In addition, the recently discovered archaea-type MVA pathway is supposed to be suitable for anaerobic life owing to the lesser consumption of energy and is thus inferred as the prototype for all other types, which subsequently evolved into the haloarchaea-type pathway, followed by the eukaryote-type and the *Thermoplasma*-type, respectively [19,25]. Therefore, the debatable basal phylogenic position of CPR in the bacterial tree of life and the emergence of the putative prototype for the MVA pathway make the origin and evolutionary history of the MVA pathway remain controversial.

Recently, the rapid advancements in DNA sequencing and bioinformatic tools have boosted the phylogenetic diversity of archaea, revolutionizing our comprehension of the origin and evolution of life. Specifically, the discovery of the phylum *Asgardarchaeota* challenges the three-domain tree of life, as it represents the closest archaeal relatives to eukaryotes and very likely bears the origin of eukaryotes [26,27,28,29]. Although the presence of the eukaryote-type MVA pathway in a few Asagrd metagenome-assembled genomes (MAGs) has been reported [30], a comprehensive understanding of the MVA pathway in Asgard archaea remains unclear and necessitates further investigation.

In this study, we focus on the distribution and evolution of the MVA pathway in the domain Archaea, especially Asgard archaea, based on the comprehensive phylogenetic analysis of key marker genes for each of the four types of MVA pathway. Our results suggest a potential archaeal origin of the MVA pathway, likely in the form of the archaea-type pathway, rather than the LUCA origin as previously proposed. Notably, phylogenetic inferences suggest that the eukaryote-type MVA pathway very likely emerged from Asgard archaea *Heimdallarchaeia* and was then transferred to the eukaryotic ancestor during eukaryogenesis, thus providing insightful clues to the two-domain tree of life. In addition, the *Thermoplasma*-type MVA pathway is also inferred to originate from the Asgard archaea based on the phylogenetic analysis, indicating the key roles of Asgard archaea in the evolution of the MVA pathway.

## 2. Materials and Methods

### 2.1. Dataset Construction

The representative archaeal genome dataset was constructed by systematically selecting at least one genome or MAG from each order based on the Genome Taxonomy Database (GTDB) Release 07-RS207. For further expanding the taxonomic diversity of Asgard archaea, here we additionally collected those novel Asgard MAGs from the latest studies [27,31,32] and the updated NCBI Genebank (final data collection in April 2023), as they had not been documented in the GTDB Release 07-RS207 at that time. The taxonomy of these MAGs was also assigned using the GTDB-Tk v1.5.1 [33]. In total, this dataset consists of 229 representative archaeal complete genomes/MAGs, which are initially categorized into the Euryarchaeota group, phylum *Thermoproteota*, phylum *Asgardarchaeota*, and the DPANN group and subsequently assigned into 18 phyla, 52 classes, and 138 orders according to the GTDB Taxonomy (r207) (Appendix A). These representative genome assemblies were predicted and translated into amino acid sequences using Prodigal v2.6.3 [34]. The genome quality was assessed using checkM v1.1.3 [35]. To guarantee the quality of the dataset, these complete genomes and high-quality MAGs (completeness ≥95%, contamination < 5%) were included in further analysis. In addition, for covering all of the archaeal taxonomic units at the order level, middle-quality MAGs (50% ≤ completeness < 90%, contamination < 10%) were also included if the order did not contain any high-quality representative genome.

### 2.2. Sequences Acquisition

The experimentally validated sequences involved in the MVA pathway were employed as the query reference to retrieve the target homologous genes in the dataset (Appendix A). Homology searches were first performed by BLASTP v2.12.0 [36] to compare those querying reference sequences against the genome dataset with the E-value < 1e^−15^ and coverage >70%. Further, a preliminary phylogenetic tree for each enzyme was constructed by using FastTree 2 with default parameters [37] to check and remove those anomalous sequences.

GHMP kinases belong to a large family with many members and share homology with each other, including not only MVK and phosphomevalonate kinase (PMK) in the MVA pathway but also homoserine kinase (HK) and galactokinase (GLK), which are irrelevant to the MVA pathway [38]. Sequences assigned to HK and GLK were also removed by phylogenetic analysis. In addition, the genomic neighborhood was also utilized for the recognition, considering that genes encoding the MVA pathway usually form gene clusters. As for homologous GHMP family decarboxylases, haloarchaea-type phosphomevalonate decarboxylase (PMD) and eukaryote-type diphosphomevalonate decarboxylase (DMD) are often misannotated in the automated annotation workflow [39]. The conserved key residues were identified after alignment (displayed by WebLogo 3 [40]) and a phylogenetic tree was constructed (see Section 2.3) to differentiate potential wrong lineages. In addition, the genome context was also used to distinguish among the homologous decarboxylases. For example, the homologs of the GHMP family decarboxylase from *Lokiarchaeia* shared homology with both PMD and *Thermoplasma*-type mevalonate 3,5-bisphosphate decarboxylase (MBD), in which the latter even had a relatively higher identity than the former; however, the other *Thermoplasma*-type enzymes, which are mevalonate 3-kinase (M3K) and mevalonate 3-phosphate kinase (M3PK), were absent in the MAGs of *Lokiarchaeia*. Hence, these decarboxylases should function as PMD rather than MBD.

To exclude the possibility of these sequences coming from metagenomic contamination or assembly artefacts, here we prioritized these Asgard MAGs of high quality (completeness >90% and contamination <5%) in the dataset construction (Appendix A) and further checked the taxonomic origin of adjacent genes on the contig carrying the key genes of eukaryote-type or *Thermoplasma*-type MVA pathway through BLASTP querying against the NCBI NR database (Appendix A).

### 2.3. Phylogenetic Analysis

To investigate the origin of the MVA pathway, the common enzymes shared by the archaeal and bacterial MVA pathways including HMGR2, MVK, and IDI2 were utilized for the phylogenetic analysis. The homologous GHMP family kinases (MVK/PMK) and the homologous decarboxylases and special kinase (PMD/DMD/MBD and M3K) are two groups of homologous enzymes that are key to revealing the evolution of the MVA pathway. Hence, the phylogenetic analysis of these homologous enzymes was performed together. In addition, the non-GHMP family enzymes in the archaea-type (AcnX1 and AMPD) and *Thermoplasma*-type MVA pathways (M3PK) were also applied to the phylogenetic analysis. In addition, coding sequences for the MVA pathway in bacteria and eukaryotes were collected from previous research [10]. The above phylogenetic analyses were performed using the following procedures: The sequences were aligned using MAFFT version 7.310 with the L-INS-i option [41]. The alignments were trimmed using TrimAl v1.4.rev15 with the automated1 option [42]. The trimmed alignments were used in subsequent phylogenetic analyses using the Maximum Likelihood (ML) method. The ML trees were constructed via IQ-TREE 2 version 2.1.2 [43] in which the best evolutionary models (LG+C60) were tested by using ModelFinder [44] with the –m MFP option and determined based on the Bayesian Information Criterion (BIC) (Appendix A). The value of branch support was evaluated using ultrafast bootstrap method (UFBoot) and Shimodaira-Hasegawa approximate likelihood ratio test (SH-aLRT) with 1000 iterations, respectively. The trees were visualized by using tvBOT [45].

## 3. Results and Discussion

### 3.1. Distribution Pattern of MVA Pathway in the Domain Archaea

In this study, a taxon-rich dataset of 229 archaeal genomes/metagenome-assembled genomes (MAGs) was collected and refined from the Genome Taxonomy Database (GTDB), which covered all of the 138 order-level taxa across 18 archaeal phyla based on the GTDB taxonomy R207. For the sake of clarity and easy comparison, we retained the conventional names of Euryarchaeota and DPANN (an archaeal superphylum including *Diapherotrites*, *Parvarchaeota*, *Aenigmarchaeota*, *Nanoarchaeota*, *Nanohaloarchaeota*, and several other phyla, assigned in the NCBI taxonomy), while phyla *Thermoproteota* and *Asgardarchaeota* were used following the GTDB taxonomic assignment. In each archaeal genome/MAG, the homologous genes for 16 MVA pathway-specific enzymes were carefully examined and mapped (Figure 1b). In total, the gene repertoires for four types of MVA pathways were identified in 16 of 19 archaeal phyla, except for some phyla from the DPANN group (Figure 1b), suggesting this isoprenoid biosynthesis pathway predominates in the domain Archaea and has an ancient evolutionary history.

The taxonomic distributions of each type of MVA pathway in archaea are reflected by the presence of their specific key enzymes in the dataset. The archaea-type MVA pathway that exclusively encodes mevalonate 5-phosphate dehydratase (encoded by AcnX1 and AcnX2) and trans-Anhydromevalonate 5-phosphate decarboxylase (AMPD) was found as the dominant isoprenoid biosynthesis pathway in the domain Archaea, given the ubiquitous distribution of its key enzymes in the majority of archaeal genomes/MAGs (Figure 1b). For the eukaryote-type MVA pathway, its two key enzymes, the phosphomevalonate kinase (PMK) and diphosphomevalonate decarboxylase (DMD), were found not only in the thermoacidophilic archaeon *Sulfolobales* as reported previously [8], but also in the phylum *Asgardarchaeota*, including classes *Heimdallarchaeia*, *Lokiarchaeia*, as well as several DPANN MAGs (Figure 1b). The order *Sulfolobales* was considered as the sole archaeal taxon that encodes the eukaryote-type MVA pathway as a consequence of HGT [10]. Here the finding of the eukaryote-type MVA pathway in broader archaeal taxa, in particular Asgard archaea, not only suggested a putative evolutionary history of this pathway in the domain Archaea but also implied the potential role Asgard archaea played in its origin and evolution.

The haloarchaea-type MVA pathway was considered as a specialized isoprenoid biosynthesis pathway for the halophilic and thermophilic archaea (class *Halobacteria* and phylum *Thermoplasmatota*) [13]. Here, our results remarkably expanded the taxonomic distribution of this MVA variant outside the Euryarchaeota group by finding its characteristic enzyme, phosphomevalonate decarboxylase (PMD), in classes *Bathyarchaeia, Nitrososphaeria*, and *Nitrososphaeria_A* of phylum *Thermoproteota* and class *Lokiarchaeia* of phylum *Asgardarchaeota* (Figure 1b). It means that the evolution of the haloarchaea-type MVA pathway is likely more complicated than we thought before. Similarly, the *Thermoplasma*-type MVA pathway has been exclusively discovered in the order *Thermoplasmata* as a modified route for adapting in extreme acidic environments [17]. Homologs of its key enzymes, including mevalonate 3-kinase (M3K), mevalonate 3-phosphate kinase (M3PK), and mevalonate 3,5-bisphosphate decarboxylase (MBD), were identified in several MAGs of the DPANN group and phylum *Asgardarchaeaota* (class *Heimdallarchaeia*) that are from non-acidic environments, implying the evolutionary driving force of this modified MVA pathway is beyond acidic adaption.

### 3.2. The Archaeal Origin of the MVA Pathway

To further decipher the origin of the MVA pathway, here we performed phylogenetic analysis of the common enzymes shared by archaeal and bacterial MVA pathways. Among them, MVK is shared by three out of the four types of MVA pathway (except the minor *Thermoplasma*-type) and is distributed in all domains of life; thus, it was selected as the characteristic enzyme to study the evolutionary history of the MVA pathway. Although a few HGT events still occurred, the majority of the MVK sequences from phylum *Thermoproteota*, Euryarchaeota group, and those deep-branching Asgard lineages including classes *Baldrarchaeia*, *Jordarchaeia*, and *Wukongarchaeia* [46], generally formed monophyletic clades (Figure 2). The topology of the MVK tree was highly congruent with the archaeal species tree [47], indicating the phylogenetic conservation and an ancient origin of this enzyme, as well as the MVA pathway in the domain Archaea. However, for those Asgard lineages that have been proposed to have relatively close phylogenies to eukaryotes, particularly the class *Heimdallarchaeia* (28), their MVK homologs did not cluster with other Asgard lineages but instead formed a miscellaneous clade interweaving with sequences from bacteria, eukaryotes, DPANN archaea, and a few other archaea (Figure 2). Similar topologies were also observed in the trees of HMGR2 and IDI2 (Appendix A). Consequently, this implies that these Asgard lineages have a complex evolutionary history of the MVA pathway, and probably contribute to the transfer of the MVA pathway to bacteria and eukaryotes.

Moreover, the majority of the archaeal lineages use AcnX1, AcnX2 and AMPD to synthesize isoprenoids through the archaea-type MVA pathway (Figure 1a), which was recently discovered and proposed as the evolutionary prototype for the three other types [19,25]. Here, our comprehensive investigation further confirmed that the archaea-type is the most universal MVA pathway in archaea, including the majority of DPANN and Asgard archaea (Figure 1b). For the majority members of the Euryarchaeota group and the phylum *Thermoproteota*, their AMPD and AcnX1 homologs have relatively conserved phylogenies on the tree (Appendix A), indicating the ancient emergence of the archaea-type MVA pathway that might be sourced back to the common ancestor of archaea.

As isoprenoids are indispensable components for all cellular life, the MVA pathway for their precursor biosynthesis was initially inferred to have a rather ancient origin, potentially as old as the LUCA (3, 7, 8, 17, 18). This inference was mainly proposed based on the specific homologs of the eukaryote-type MVA pathway found in CPR bacteria (18), which was considered to be placed at the base of the domain Bacteria. However, the *CPR* bacteria recently have been placed as a sister lineage to the phylum *Chloroflexota*, instead of a deep branch diverged from the bacterial root (19, 20), thus challenging the hypothesis of an ancient emergence of the MVA pathway in the last bacterial common ancestor (LBCA). In our study, phylogenic analysis results suggested a horizontal origin of the MVA pathway found in bacteria and eukaryotes, therefore arguing against the previous hypothesis of the presence of the MVA pathway in the LBCA or LUCA. Moreover, Asgard archaea are inferred to play a key role in driving the evolution of the MVA pathway, probably as the potential donor of this isoprenoid biosynthesis pathway to bacteria and eukaryotes, which is supported by the enriched integrons recently discovered in Asgard MAGs [48] and their close phylogenies to eukaryotes (28). In addition, the newly discovered archaea-type MVA pathway was proposed as the evolutionary prototype due to its lower energy requirement that might facilitate the anaerobic metabolism for the early evolution of life (16, 21). Here, our results also suggest an early origin of the archaea-type pathway by finding its ubiquitous distribution and phylogenetic conservation of key enzymes in current archaeal lineages. In conclusion, the MVA pathway, very likely in the form of the archaea-type pathway, is proposed to have an ancient origin within the domain Archaea, not the LUCA inferred previously, which subsequently evolved into other types and was transferred to bacteria and eukaryotes with great contribution from the Asgard archaea.

The lipid divide is an interesting phenomenon that depicts the differentiation of membrane lipids among archaea, bacteria, and eukaryotes [49,50]. Given the pivotal role of isoprenoids in synthesizing archaeal membrane lipids, elucidating the origin of the MVA pathway is crucial for understanding the lipid divide between archaea and bacteria, which contributes to our comprehension of the LUCA [50,51]. In this study, multiple lines of evidence support an archaeal origin of the MVA pathway, while the MEP pathway, the other non-homologous pathway for isoprenoid biosynthesis, is highly conserved in bacteria and almost absent in archaea [52] and is thus inferred to originate in the last bacterial common ancestor (LBCA). Considering the currently accepted knowledge of a simple and small gene set in the LUCA [53], it seems unreasonable that the LUCA possesses two highly specialized redundant metabolic pathways for isoprenoid biosynthesis, but then selectively retained one of each after the divergence of archaea and bacteria. Therefore, the MVA and MEP pathway for isoprenoid biosynthesis very likely have independent origins from the last archaeal common ancestor (LACA) and LBCA, respectively, after their divergence from the LUCA. If this scenario holds true, it is theoretically expected that the LUCA would lack both of the two non-homologous pathways for isoprenoid biosynthesis, indicating the enzymatic incapability of the LUCA to synthesize isoprenoids required for archaea-type membrane lipids. Regarding other aspects of archaea-type membrane lipids, the enzyme responsible for synthesizing G1P lipids is thought to be present in the LUCA. It is proposed that the cell membrane represents a fundamental characteristic that directly reflects the evolutionary stage of cellularity [54]. The incompleteness of the biosynthesis in the LUCA may reflect that the biosynthesis of membrane lipids could be still rapidly evolving, implying a progenotic LUCA rather than cellular entity (genotes) [54,55]. Moreover, several recent studies demonstrated the stable self-assembly of single-chain ampliphiles under hydrothermal conditions, implying the possible existence of a fatty acid in the membrane of the protocell [56,57]. While the prebiotic synthesis of isoprenoids in nature remains unclear, isoprenoids have been demonstrated to enhance the stability of fatty acid membranes; however, a high concentration of isoprenoid substrates is requisite for the incorporation of isoprenoids into fatty acid membranes [58]. Considering the high demand for isoprenoids, the emergence of the MVA pathway may have enabled the ancestor of archaea to autonomously synthesize isoprenoids, thereby facilitating the incorporation of isoprenoids into fatty acid membrane lipids and enhancing the membrane stability. In this scenario, the archaeal origin of the MVA pathway may play a pivotal role in driving the emergence of archaea-type cell membranes. In conclusion, our results provide insights into the physiological status and lipid divide of the LUCA in terms of isoprenoids’ biosynthesis. More convincing evidence through further study, in particular study of synthetic biology, is essential to address this question.

### 3.3. The Evolution of the Haloarchaea-Type and Thermoplasma-Type MVA Pathway

The haloarchaea-type MVA pathway is thought to have evolved relatively late from the archaea-type pathway [19,25] due to its limited distribution in the aerobic halophiles from the class *Haloarchaeia* and some *Chloroflexota* [12,13]. Here, our work remarkably expanded the distribution of the haloarchaea-type MVA pathway in the domain Archaea by finding its key enzyme PMD in phyla *Thermoplasmatota*, *Thermoproteota*, and *Asgardarchaeota* (Figure 1b), along with rather conserved phylogeny on the trees (Appendix A). These results indicate that the haloarchaea-type MVA pathway might also have an ancient origin in the domain Archaea, probably as early as the common ancestor of the Euryarchaeota group and phylum *Thermoproteota*, which is much older than we thought before. Moreover, its key enzyme PMD has been demonstrated to consume one more molecule ATP to synthesize IPP or DMAPP compared to the archaea-type pathway, which thus was inferred more suitable for the aerobic haloarchaea in extreme environments (16). However, here we also found that the haloarchaea-type route is the sole MVA pathway for diverse strictly anaerobic archaea, including the isolated Asgard archaeon Candidatus *Prometheoarchaeum syntrophicum* MK-D1 [59], thus supporting a plausible early presence of the haloarchaea-type MVA pathway in the domain Archaea, probably not much later than the primordial archaea-type pathway.

The complete *Thermoplasma*-type MVA pathway was firstly identified in the order *Thermoplasmatales* of phylum *Thermoplasmatota*. In this study, the key enzymes of this type, namely M3K, M3PK, and MBD, were also discovered in several lineages of phylum *Asgardarchaeota* and the DPANN group (Figure 1b), indicating their important roles in the evolution of this special MVA pathway. Based on the phylogenetic analysis, all of M3K, M3PK, and MBD display similar topologies on the tree (Figure 3a–c): sequences from *Thermoplasmatales* are placed closed to the class *Heimdallarchaeia* (*o_ JABLTI01*) as sister groups, together with several DPANN lineages, including phylum *Micarchaeota*, *Altiarchaeota*, and *Aenigmatarchaeota*, indicating that the *Thermoplasma*-type MVA pathway might have evolved within the class *Heimdallarchaeia* or its ancestral lineages and was latterly transferred to *Thermoplasmatales* via HGT. Moreover, consistent with a previous study that reported that the MBD might have evolved from the homologous PMD of the haloarchaea-type [25], here our results suggest that all of the MBD sequences derived from *Thermoplasmatales* form a sister group to the PMD of *Lokiarchaeia*, which further supports the close relationship between *Asgardarchaeota* and *Thermoplasmatales* in the evolution of the MVA pathway.

The *Thermoplasma*-type pathway has been confirmed to produce isoprenoids efficiently under low pH conditions, which might be a specialized metabolic strategy for the acidophilic *Thermoplasmatales* to survive in extremely acidic environments [17]. The *Thermoplasma*-type MVA pathway is thought to evolve from the haloarchaea-type pathway through the evolution from PMD to MBD and M3K, accompanied by the loss of MVK and the emergence of M3PK [25]. In this study, the phylogenies of M3K, M3PK, and MBD all support the relationships between *Asgardarchaeota* and the *Thermoplasma*-type MVA pathway. In addition, the homologs for the *Thermoplasma*-type MVA pathway were identified in some MAGs of the DPANN phyla *Micarchaeota* and *Aenigmatarchaeota* (Figure 1b). Among them, the gene repertoire of the *Micarchaeota* MAGs was acquired horizontally from their host *Thermoplasmatales* as previously reported [18], while the phylum *Aenigmatarchaeota* is inferred to mediate the horizontal transfer of the *Thermoplasma*-type MVA pathway from the Asgard archaea to the *Thermoplasmatales*. It is worth mentioning that the MAGs containing genes encoding the *Thermoplasma*-type pathway also harbor the eukaryote-type PMK and DMD. The mixed combination of the two types may implicate a more complex history of the MVA pathway than Aoki’s hypothesis [25].

### 3.4. The Origin and Evolution of the Eukaryote-Type MVA Pathway

With the inclusion of more Asgard MAGs, the homologous genes encoding the key enzymes for the eukaryote-type MVA pathway, namely PMK and DMD, were extensively detected in classes *Lokiarchaeia* and *Heimdallarchaeia* (Figure 1b). Among them, most MAGs from the class *Heimdallarchaeia* (including orders *Hodarchaeales* and *JABLTI01*) have both of the key genes, while those from order *CR-4* (class *Lokiarchaeia*) only possess the DMD homologs but lack the PMK (Appendix A). In addition, the order *Hodarchaeales* was proposed as the closest archaeal relative to eukaryotes [46]; here, we found that their MAGs not only possess both PMK and DMD but also encode the classical archaea-type marker genes (AMPD and AcnX) on the same contig (Appendix A), indicating these Asgard taxa play key roles in the evolution of the eukaryote-type MVA pathway (Figure 4a) and potentially act as the archaeal donor of this isoprenoid biosynthesis metabolism to the eukaryotic ancestor.

It is postulated that the key enzymes PMK and DMD for the eukaryote-type MVA pathway originated from highly homologous MVK and PMD for the haloarchaea-type pathway, respectively [25]. By using the MVK and PMD as outgroups, our phylogenetic analysis showed that PMK sequences from order *JABLTI01* and DMD sequences from order *CR-4* formed the closest clade adjacent to the outgroups on the tree, respectively (Figure 4a,b). Moreover, DMD from *Chloroflexota* has been demonstrated to process additional PMD activity for catalyzing the MVA-5P to IP [39]. The DMD homologs from order *CR_4* exhibit a strikingly similar motif pattern to those found in *Chloroflexota*, as both lack the conserved lysine residue of DMD but retain the conserved histidine residue of PMD (Appendix A). Therefore, it is highly likely that these DMD homologs from order *CR_4* also possess the dual enzymatic specificity observed in *Chloroflexota*, implying that these sequences might be at the intermediate stage of evolution from PMD to DMD. These results suggest that the enzymatic evolution of the eukaryote-type MVA pathway very likely occurred within the phylum *Asgardarchaeota*, potentially in the classes *Lokiarchaeia* and *Heimdallarchaeia* separately, or their common ancestor. In addition, the sequences of the eukaryote-type MVA pathway from other lineages of Asgard archaea were generally intertwined with sequences from bacteria (including CPR) and the DPANN group on the trees (Figure 4a,b), implying that multiple HGT events involved in the eukaryote-type MVA pathway have occurred independently among bacteria, eukaryotes, and *Sulfolobales* after the initial emergence from the Asgard archaea.

One of the unresolved issues in eukaryogenesis is the transformation of the eukaryotic cell membrane from archaeal to eukaryotic/bacterial lipids. Several studies have reported the presence of genes involved in fatty acid and ester-bond biosynthesis of the bacterial/eukaryote type in some Asgard MAGs, implying that Asgard archaea may possess chimeric membrane lipids containing both eukaryotic/bacterial and archaeal lipids, which might be the key to illustrate the lipid divide between eukaryotes and archaea [60,61]. In this study, we found that some Asgard lineages, especially the class *Heimdallarchaeia* that has close phylogeny to eukaryotes, possess the eukaryote-type MVA pathway, which was demonstrated to be less effective than the constructed haloarchaea-type MVA pathway in yeast [62]. Therefore, the putative evolution of the inefficient eukaryote-type MVA pathway from the haloarchaea-type pathway might reflect the possibility that these Asgard archaea have reduced demand for isoprenoid synthesis, thus implying that they may process the mixed membrane with both isoprenoids and fatty acids. This hypothesis of mixed membrane lipids in Asgard archaea is crucial to understanding the lipid divide between archaea and eukaryotes, which is inferred to occur within certain ancestral Asgard archaea as indicated by our current results. It is imperative for future research to comprehensively address this issue by investigating the membrane lipid composition of Asgard archaea.

In summary, contrary to the previous assumption that the eukaryote-type MVA pathway existed in the ancestor of bacteria or even the LUCA [10], here, our results provide support for the origin of the eukaryote-type pathway within Asgard archaea, followed by its horizontal transfer into bacteria and potential contribution to the eukaryotic MVA pathway. Not only does this work provide unique insight into the evolutionary relationship between Asgard archaea and eukaryotes in terms of lipid biosynthesis, but it also reinforces the key roles of Asgard archaea in eukaryogenesis, thus supporting the two-domain tree of life scenario.

## 4. Conclusions

In conclusion, our findings suggest that the MVA pathway originated from archaea rather than the LUCA as previously assumed (Figure 5), indicating the enzymatic incapability of the LUCA to synthesize isoprenoids and the key roles of the MVA pathway in the divergence of archaea from the LUCA. Significantly, the identification of the mosaic MVA pathway, coupled with subsequent phylogenetic analysis, underscores the pivotal roles played by Asgard archaea in the evolution of MVA pathway. Specifically, it is proposed that the *Thermoplasma*- and eukaryotes-type MVA pathway originated from the class *Hemidallarchaeia*, and the latter is inferred to be transferred to the eukaryotic ancestor during eukaryogenesis (Figure 5). Further investigations into the membrane lipids of Asgard archaea are imperative for a comprehensive understanding of the early evolution of eukaryotes.

## Figures and Tables

**Figure 1 microorganisms-12-00707-f001:**
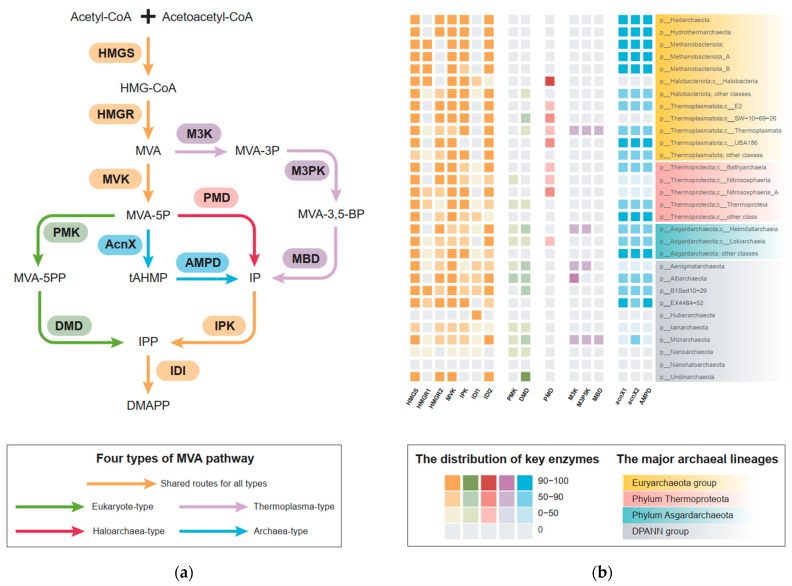
Four types of MVA pathway discovered and their distribution in the domain Archaea. (**a**) The enzymes within identically colored boxes function in the same type of MVA pathway, including the shared route. Abbreviations for enzymes: HMGS, 3-hydroxy-3-methylglutaryl-CoA synthase; HMGR, 3-hydroxy-3-methylglutaryl-CoA reductase; MVK, mevalonate kinase; PMK, phosphomevalonate kinase; DMD, Diphosphomevalonate decarboxylase; PMD, phosphomevalonate decarboxylase; M3PK, mevalonate 3-phosphate 5-kinase; MBD, mevalonate 3,5-bisphosphate decarboxylase; AcnX, MVA5P Dehydratase; AMPD, tAHMP Decarboxylase; IPK, isopentenyl phosphate kinase; M3K, mevalonate 3-kinase; IDI, IPP/DMAPP isomerase; (**b**) The heatmap shows the prevalence of genes encoded key enzymes of four types of MVA pathway in the representative MAGs from different archaeal phyla or classes. The archaeal taxon in each row are distinguished by a different background color: DPANN group in gray, phylum *Asgardarchaeota* in cyan, phylum *Thermoproteota* in red, and Euryarchaeota group in yellow. The color of the cell represents the type of MVA pathway, where common enzymes in MVA pathway are in gray, eukaryote-type enzymes are in green, haloarchaea-type enzyme is in red, *Thermoplasma*-type enzymes are in purple, and archaea-type enzymes are in blue. The shade of a cell’s color indicates the proportion of MAGs encoding this gene in the total number of representative MAGs from this archaeal taxon (Appendix A).

**Figure 2 microorganisms-12-00707-f002:**
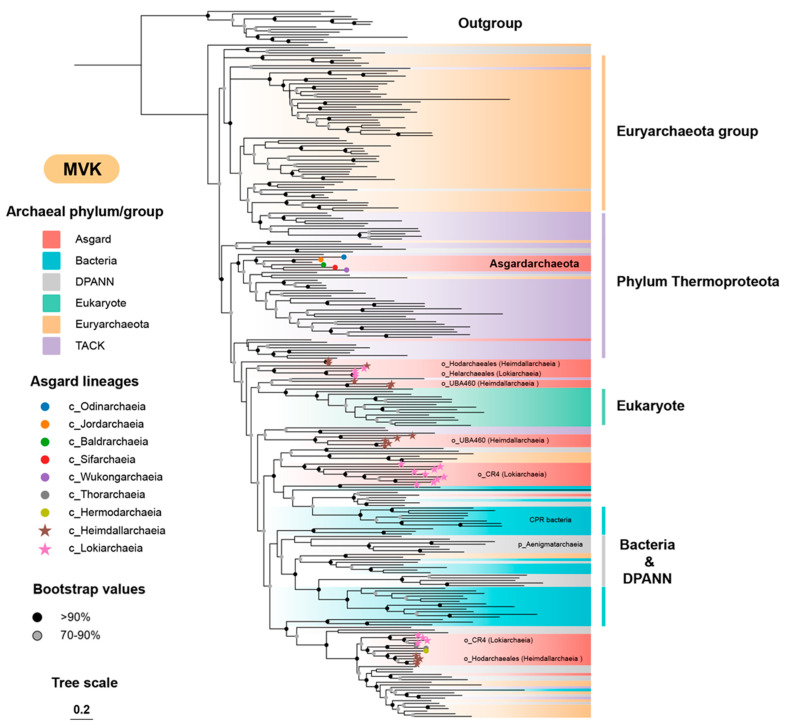
The phylogenetic tree of MVK. The maximum-likelihood tree was constructed based on 260 amino acid sequences using IQTREE2 with the LG+R8+C60 model. Homologous GHMP family enzyme galactokinases were used as the outgroup.

**Figure 3 microorganisms-12-00707-f003:**
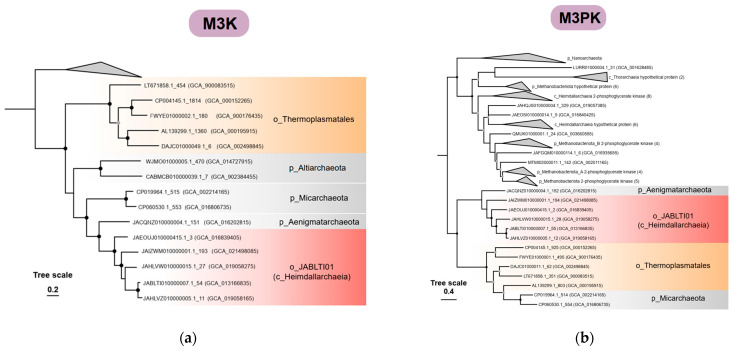
The phylogenetic trees of kinases and decarboxylases in the *Thermoplasmata*-type MVA pathway; (**a**) The maximum likelihood tree of mevalonate 3-kinase (M3K) was constructed based on the 15 amino acid sequences using IQTREE2 with the LG+I+G4+C60 model. DMD was selected as the outgroup; (**b**) The maximum likelihood tree of M3PK was constructed based on the 14 amino acid sequences using IQTREE2 with the LG+I+G4+C60 model. The outgroup was selected as 2-phosphoglycerate kinase and some other homologous sequences with unknown function; (**c**) The maximum likelihood tree of MBD was constructed based on the 7 amino acid sequences using IQTREE2 with the LG+I+G4+C60 model. Homologous PMD sequences were selected as the outgroup.

**Figure 4 microorganisms-12-00707-f004:**
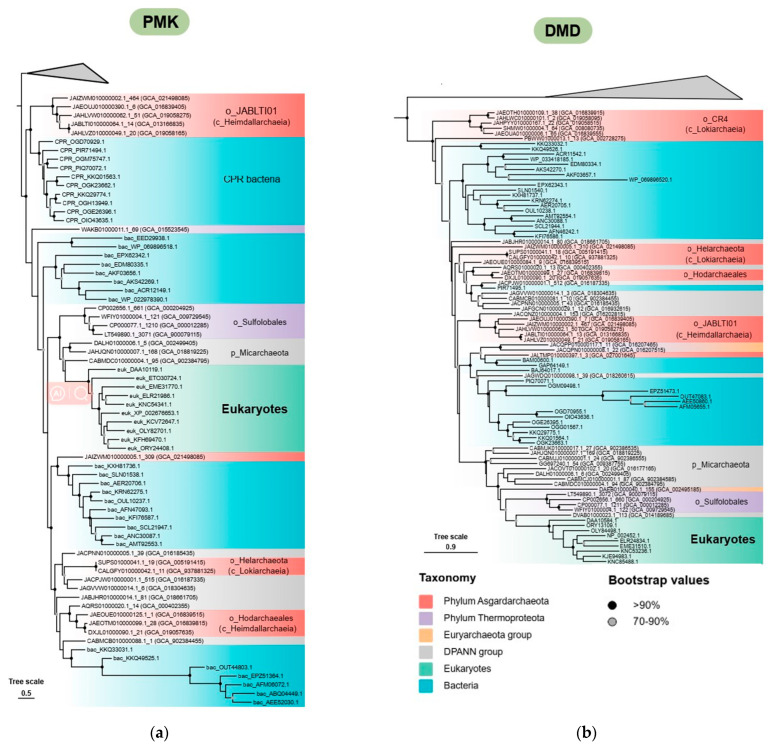
The phylogenetic trees of kinases and decarboxylases in the eukaryote-type MVA pathway; (**a**) The maximum likelihood tree of GHMP family phosphomevalonate kinase (PMK) was constructed based on 71 PMK sequences and 111 MVK sequences using IQTREE2 with the LG+R8+C60 model; (**b**) The phylogenetic tree of GHMP family diphosphomevalonate decarboxylase (DMD) was constructed based on 87 DMD sequences and 29 PMD sequences using IQTREE2 with the LG+R8+C60 model.

**Figure 5 microorganisms-12-00707-f005:**
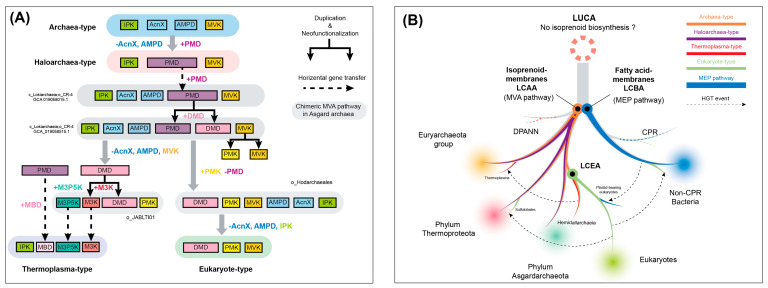
Hypotheses for the evolution of the MVA pathway (**A**) The hypothetical scheme of molecular evolution for the four types of MVA pathway. Colored boxes indicate the enzymes used in four types of MVA pathway. +/- and dash lines represent the inferred gene duplication, loss, and horizontal transfer events for specific genes, respectively. Those enzyme arrays with gray background represent those chimeric MVA pathways identified in different Asgardarchaeota MAGs; (**B**) The proposed scenario for the evolutionary history of isoprenoid biosynthesis pathway in Earth life. The archaea- and haloarchaea-type MVA pathways are inferred to originate from the last common archaeal ancestor (LCAA). The eukaryote- and Thermoplasma-type MVA pathways likely emerged within the order Hemidallarchaeia of phylum Asgardarchaeota, in which the former might be vertically transferred to the last common eukaryotic ancestor (LCEA) during eukaryogenesis, while the latter was probably acquired by the Thermoplasma lineage via HGT. The dashed lines indicate the inferred HGT events of the MVA pathway occurred among different archaeal major lineages, or across bacteria and eukaryotes. In addition, the MEP pathway is generally regarded to emerge from the last common bacterial ancestor (LCBA); therefore, it is more plausible that neither the MVA or EMP pathway was possessed by the last universal common ancestor (LUCA), implying that the LUCA might have no membrane or a non-cellular structure.

## Data Availability

Data are contained within the article and Appendix A.

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
