# Peer review of "Expanded Archaeal Genomes Shed New Light on the Evolution of Isoprenoid Biosynthesis"

_microorganisms, 2024, doi:10.3390/microorganisms12040707_

Round 1
Reviewer 1 Report
Comments and Suggestions for Authors
The MS submitted by Zhu summarises and study on archaeal genomes to shed light on the evolution of isoprenoid biosynthesis. The topic is of interest and fits the scope of the journal. It is well written, however, there is a kind of imbalance in terms of representative genomes of almost all archaeal phenotypes, which is the main concern of this work. On the other hand, the labels in the figures can not be read. Besides, some of the most recent works on MVA in archaea have not been discussed, mainly works related to haloarchaea.
Consequently, to improve the work, the following issue should be considered:
- To increase the number of archaeal genomes/MAGs to get a significant and representative analysis. it would be nice to see genomes of archaeal species from which details on biochemistry, microbiology and physiology related to this pathway are available in the literature. Only 211 genomes have been considered representatives of the whole Archaea Domain. In works just focused on haloarchaea, a similar number of genomes were considered.
Apart from minor issues related to typos and grammar, two more minor comments have been embedded through the MS to help the authors improve this version.

Comments on the Quality of English LanguageMinor issues related to typos and grammar
Author Response
“however, there is a kind of imbalance in terms of representative genomes of almost all archaeal phenotypes, which is the main concern of this work…Consequently, to improve the work, the following issue should be considered: To increase the number of archaeal genomes/MAGs to get a significant and representative analysis… Only 211 genomes have been considered representatives of the whole Archaea Domain. In works just focused on haloarchaea, a similar number of genomes were considered.”
RE: Yes we understand the reviewer’s concern about the number of archaeal genomes/MAGs used in this study might not be representative enough. However we’re confident the current genomes are enough for our study based on following reasons:
(1) The selection of number of MAGs depends on the purpose of studies, for instance, one may select more than hundreds of MAGs for even only one order of archaea to study its evolution. As for our case, the aim of this study is to delineate the deep origin and evolution history of the MVA pathway in the domain Archaea, therefore our analysis should cover archaeal taxonomy as broad as possible. The current genomes/MAGs used in the study covered all the 132 orders of Archaea in the archaeal GTDB taxonomy (RS207), ensuring each order has at least one (mostly five) representative genome/MAG (Table S1).
(2) With the two exceptions at family level (eukaryote-type MVA pathway in Sulfolobales and Thermoplasma-type in Thermoplasmatales), the difference of distribution pattern of MVA pathway for the majority of archaea is at the class level (Figure 1b and Table S5). Therefore,genomes chosen at the resolution of the order-level in archaeal taxonomy is sufficient to describe the distribution pattern of MVA pathways in archaea.
(3) The MVA pathway in many archaeal lineages has been studied in detail in previous studies (Lombard et al., 2010; Hoshino et al., 2018; Hayakawa et al., 2018). Hence, our analysis specially controlled the number of genomes/MAGs of well-studied archaeal lineages such as Haloarchaea. On the other hand, as the MVA pathway of those newly discovered archaeal lineages are still poorly reported, here we increased the number of genomes/MAGs of previously unstudied archaeal lineages, especially Asgard archaea, that is also the main highlight of this study.
To sum up, we are confident that the number of genomes/MAGs we used in this study is sufficiently representative to support our analysis and conclusion.
“On the other hand, the labels in the figures can not be read. ”
RE: Figures with clear labels are used (Fig 3).
“Besides, some of the most recent works on MVA in archaea have not been discussed, mainly works related to haloarchaea.”
RE: Thanks for your advice. We have noted the relevant literatures on MVA pathway in halophilic archaea you mentioned and have added it in the introduction section (line 49-55)
“Minor issues related to typos and grammar”
RE: Thanks, we have checked and corrected the typos, and polished the grammar.

Reviewer 2 Report
Comments and Suggestions for Authors
This is a very interesting manuscript, well-edited. I would like to ask for the following revision. Since according to the authors the MVA pathway originated in Archaea rather than LUCA, then the authors must explicitly say that LUCA was a progenote and not a cellular entity. That is, it seems to me that in the manuscript little importance is given to this very important consideration. After this review, the manuscript could be accepted.
Author Response
"Since according to the authors the MVA pathway originated in Archaea rather than LUCA, then the authors must explicitly say that LUCA was a progenote and not a cellular entity. That is, it seems to me that in the manuscript little importance is given to this very important consideration."
RE: Thank you for your valuable advice! As you suggested, the late origin of the MVA pathway involved in the synthesis of archaea-type membrane lipids suggests that LUCA is still in a still rapid and progressive evolution, reflecting the progenotic stage of LUCA. We have highlighted this point and added relevant content in the discussion (line 310-314).

Round 2
Reviewer 1 Report
Comments and Suggestions for Authors
Thanks for your time addressing all the comments and suggestions made by this reviewer.
Comments on the Quality of English LanguageMinor grammar and typo issues.